# OpenReview forum: "Position: The Pace of AI Innovation Demands Matching Urgency in Societal Impact Research to Shape Economic Policy"
_ICML.cc/2025/Position_Paper_Track — Submitted to ICML 2025 Position Paper Track_

### Official Review · Reviewer_6hkf · 2025-02-21

**Significance:** 3
**Argument Clarity:** 3
**Rating:** 3
**Confidence:** 4

**Questions:**

To what extent does the author think corporate tech sector practices and culture may be worsening or accelerating the discussed human and labor crises, separately from the advancement of AI technology? What role do tech companies, if any, have in setting up better norms of human valuation and respect if policy makers are lagging with solutions? What role do universities have in setting better norms and expectations for computer science graduates if the number of tech sector programming jobs that are stable and higher paying may steadily decrease?

For the end paragraph, there is the sentence "This analysis points toward several crucial socio-technical research directions: understanding sector-specific automation trajectories, developing adaptive educational models, and designing policy frameworks that encourage innovation but maintain broad-based economic opportunity in an increasingly automated knowledge economy", but no citations are given. Can the authors find any recent promising references in these areas to guide early efforts?

I think it's also worth some discussion that the risks mentioned can occur even if AI does not become high performing. Humans seem to have a bias towards replacing human labor with AI prematurely and overly trusting it, some example references:

Holbrook, Colin, et al. "Overtrust in AI Recommendations About Whether or Not to Kill: Evidence from Two Human-Robot Interaction Studies." Scientific reports 14.1 (2024): 19751.

Messeri, Lisa, and M. J. Crockett. "Artificial intelligence and illusions of understanding in scientific research." Nature 627.8002 (2024): 49-58.

Narayanan, Arvind, and Sayash Kapoor. "AI snake oil: What artificial intelligence can do, what it can’t, and how to tell the difference." AI Snake Oil. Princeton University Press, 2024.

**Discussion Potential:**

4

**Paper Summary:**

Two of the quotes that captures the paper's position well are:

(1) "The rapid pace of AI innovation demands equally urgent research into societal impacts to inform policy interventions that preserve economic mobility and protect individual opportunity."

(2) "Rather than treating AI advancement and societal preparation as separate challenges, we must recognize them as fundamentally interconnected. Realizing AI’s potential benefits requires parallel investment in human capital development and robust social support systems"

The paper is concerned that the AI technology revolution is markedly different than prior technological revolutions in human history. Prior revolutions have largely increased labor productivity, and numbers of and quality of jobs, but AI is unique in its ability to replace knowledge workers, creative occupations, etc. Additionally the speed of the revolution might be much faster than prior revolutions and faster than policy makers can adjust to societal changes. The authors call upon computer scientists to work with policy makers, economists and social scientists to more urgently and comprehensively look at the changes in tech sector job duties, salaries, numbers in addition to pushing for more aggressive and continual examination of university education to make sure students aren't taking on huge debt for jobs that will soon not exist.

**Position:**

Yes

**Position In Title:**

Yes

**Related Work:**

3

**Strengths And Weaknesses:**

The paper is well-written and information. In the higher level conceptual sense, I think the general worries will be familiar to most readers about AI replacing humans, job loss, insufficient safety nets, etc. However, the author provides updated statistics and nuanced details to back up their concerns and compiles useful references within one comprehensive paper. The MLB analogy is also powerful, novel and compelling about how the shifts in job numbers, duties, automation etc. can have serious impacts on graduates of computational science programs, and thus academics need to more carefully track trends and critique their curricula to better serve students and society, similar to the challenges high school and university athletes may face with just a very small number of very high paying jobs for athletic tracks.

The authors provide a good discussion of alternative views and debates, such as the downsides of excessive regulation. The paper comes across as fair and balanced in its arguments generally, and would be productive for improving discussion in the position topics.

There are some omissions and weak points. The authors primarily focus on policy pressure and solutions, such as finding the right balance of regulation on AI companies that does not stifle them, and improving social safety nets. These are an important facet of the issue, but there is no discussion of tech/AI company internal governance, management style, the increasing number of mass layoffs and employee turnover, etc. Are these layoff waves truly just from AI lowering labor costs and getting better at programming, or are they potentially also a symptom of (arguably toxic) management styles that turn tech workplaces into hyper competitive international tournaments, the increasing reliance on gig economy labor and contractors, outsourcing, and so on.

The reference list has strange capitalization trends and should be cleaned up.

**Support:**

4

---

### Official Review · Reviewer_SFoW · 2025-03-14

**Significance:** 2
**Argument Clarity:** 2
**Rating:** 1
**Confidence:** 4

**Questions:**

1. The paper provides a survey of a large swath of work projecting the impact of AI on the economy and the labor market. The paper's position, however, seems to be that more such work is needed, or more depth of such work is needed. How do the authors reconcile these two points?

2. How is the paper's argument different from (or adding to?) the existing arguments of economists who study AI and labor, such as David Autor?

**Discussion Potential:**

3

**Paper Summary:**

This paper takes the position that given AI's rapid pace of innovation, there is need for equally urgent research into the societal impacts of its real-world use, to in turn support timely policy interventions.

**Position:**

Yes

**Position In Title:**

Yes

**Related Work:**

3

**Strengths And Weaknesses:**

Strengths:

The paper has some interesting high-level ideas, including the general discussion of AI's potential economic impacts. It provides a review of policy analyses and technical reports/whitepapers that estimate AI's potential impact to disrupt various white-collar job markets, including software engineering and law. There are also the starting points of some potentially fruitful analogies to other subfields of economics.

Weaknesses:

While there are good high-level instincts in the writing, the ideas as presented here come across as disjointed and largely disorganized. There is a section about an analogy between AI development and the the MLB model (major league baseball)/sports economics more generally, but that section of the paper then goes on to provide a review of the policy papers/whitepapers/technical analyses that focus entirely on AI's potential to disrupt the labor market in various ways. The sports economics analogy is left underbaked, and it is unclear to a reader what the purpose of it might be in the context of the overall writing/message.

Similarly, the discussion of AI replacing software engineers is backed up by estimates and numbers from a wide variety of think tanks and research orgs (including financial projections/reports from Goldman Sachs, for example), as is the discussion of AI replacing lawyers or most law-related tasks. However, once again, the paper reads like a summary of the relevant industry consensus about the subject, rather than a position paper.

I think this paper is a good summary of ideas/notes that could turn into a good position paper or review paper with quite a few more rounds of revision and addition. There are many good seeds of ideas there, but the overall argument is not cohesive or structured to present a detailed, thought-provoking position.

**Support:**

2

---

### Official Review · Reviewer_J816 · 2025-03-14

**Significance:** 3
**Argument Clarity:** 2
**Rating:** 1
**Confidence:** 4

**Questions:**

-

**Discussion Potential:**

2

**Paper Summary:**

The paper argues that AI researchers and analysts should focus on the economic transformations that the adoption of AI might bring out, and that this needs to happen more urgently, given the rapid pace of AI innovation. It provides an analogy to the winner-take-all effects in major league baseball to illustrate how this change could affect knowledge workers.

**Position:**

Yes

**Position In Title:**

Yes

**Related Work:**

2

**Strengths And Weaknesses:**

The paper identifies an interesting shortcoming of AI discourse today: there is a lack of focus on the impact of AI adoption across society. This is an important question, and one that needs to be grappled with by interdisciplinary researchers. If the researchers could defend this position, they would enable AI discourse to advance significantly and prompt urgent work in understanding the societal impact of AI.

To defend the position, the authors need to make and defend a few claims:

1. The pace of AI innovation with lead to sudden and drastic AI adoption.
2. The adoption of AI will be accompanied by rapid labor displacement.
3. Existing research on societal impacts to shape economic policy is not keeping up with the pace of AI adoption.
4. If it did, we would be able to counter the worst effects of AI-related labor displacement.

In particular, the authors must defend all four claims to defend their position successfully.

Unfortunately, in its current form, the paper fails to defend any of the four claims.

First, the history of general-purpose technologies has shown that there are decades between innovation and adoption (e.g., Jeff Ding’s book on great power competition and the role of technology). To argue the first claim, the authors need to justify why AI transformations would be different from previous general-purpose technologies. The authors try to do this by asserting, at various points, that autonomous agents could perfrom various tasks more cheaply than humans, often citing improvements on benchmark accuracy (e.g., the bar exam), as evidence. But there is a big gap between benchmarks and the real world, as has been pointed out dozens of times by the growing AI evaluation community.

Second, the authors need to defend the claim that AI adoption will lead to replacement or displacement of labor, rather than augmentation of productivity improvements. Here, once again, the paper does a cursory analysis by comparing the economic costs of using AI vs. humans, and hinting at the distribution effects based on the analogy to MLB. Unfortunately, this is far from a complete defense. There are many competing economic theories about the shape of economic transformations, and there are ongoing debates within econ about the impact of AI. The paper doesn’t engage with this debate.

Third, the authors need to defend the claim that existing research is not doing enough to understand AI’s impact on society. This, after all, is their main contention: we need to rapidly increase the urgency of social impact research. Once again, the authors don’t engage with this claim at all, and instead simply assert it to be true. But there is a large community of researchers engaging with precisely this question: economists of AI, sociologists understand the qualitative changes to professions because of AI, the FAccT community etc. The authors could make plausible defenses of this claim, but the current draft offers no engagement.

Finally, even if the paper had engaged with the three previous arguments, it still needs to argue why more research on societal impacts could avert the worst outcomes that the paper predicts. What could societal impact research accomplish, and why should we be confident in its ability to change the incentives, especially given the strong incentives to automation that the paper claims exist and the short time we have before such automation comes to fruition? Unfortunately, the paper’s arguments on this point lack internal consistency.

For all of these reasons, I don’t think the paper is able to cleanly state and defend its position. I agree that there could be a defense of the position that is clear and arguable, but unfortunately, the paper’s current content doesn’t offer that defense.

**Support:**

1

---

### Official Review · Reviewer_kpjN · 2025-03-22

**Significance:** 3
**Argument Clarity:** 1
**Rating:** 1
**Confidence:** 4

**Questions:**

No specific questions

**Discussion Potential:**

2

**Paper Summary:**

This paper explores how AI may yield concentrated markets (akin to the MLB in professional sports), posturing at some learnings from that and, more generally, pointing to the need for greater understanding of AI's societal and economic impacts.

Strengths:
- The need to strengthen the focus on the socioeconomic impacts of AI is clearly very relevant
- I found the comparison to the MLB very unexpected and intriguing.

Weaknesses:
- The comparison to the MLB is entirely unmotivated. Why is AI similar in any way to major league baseball: there are many other highly concentrated markets that could also have been considered in a case study, but more importantly, why should we expect AI to trend in this direction?
- The work does not engage with a growing body of work on the economic impacts of AI: while the work notes Clio from Anthropic, there are many other works that are not discussed, even in the past couple of years like Erik Brynjolfsson's work on doing concrete case studies of deploying GenAI in call centers, Tyna Eloundou's work at OpenAI on exposure of jobs to AI, Anton Korinek and Daron Acemoglu's respective works making conflicting predictions about AI's impact on the economy, and other works by influential economists like John Horton, David Autor, and others. Further, the work does not engage with work that does do market surveillance and other forms of post-deployment monitoring on AI and the supply chain, often motivated by market concentration risks, such as work from Rishi Bommasani on the foundation model supply chain or large-scale market surveillance by the UK competition and markets authority.
- Ultimately, while the discussion of the MLB is intriguing and unexpected, it doesn't feel adequately substantive (i.e. it doesn't seem much is learned by bringing this juxtaposition to the fore). For the most part, it seems like most of the points could be made roughly equally by just talking about the abstracted concept of concentrated/winner-take-all markets. Also, explanatory factors for the types of Matthew effect emerging in the MLB that could be useful diagnostics for early identification in AI are not explored.

**Position:**

Yes

**Position In Title:**

Yes

**Related Work:**

1

**Strengths And Weaknesses:**

See above.

**Support:**

2

---

### Decision · Program_Chairs · 2025-04-27

**Decision:**

Reject

**Comment:**

Reviewers agreed that the field needs to strengthen its focus on the socioeconomic impacts of AI. While the paper presented interesting claims and information, reviewers felt that the discussion of prior work on the economic impacts of AI was weak/did not include critical references and the evidence provided did not sufficiently support the claims made. An especially salient point was that the authors did not present evidence for how more research on socioeconomic impacts would effectively avert the negative consequences of AI’s current trajectory.